# Preliminary Evidence of the Association between Time on Buprenorphine and Cognitive Performance among Individuals with Opioid Use Disorder Maintained on Buprenorphine: A Pilot Study

**DOI:** 10.3390/ijerph20166610

**Published:** 2023-08-19

**Authors:** Irene Pericot-Valverde, Kaileigh A. Byrne, Erik G. Ortiz, Stephanie Davis, Ethan Hammond, Shadi Nahvi, James F. Thrasher, Laksika B. Sivaraj, Sam Cumby, Eli Goodwin, Ashley C. King, Julia Arnsten, Sergio Fernández-Artamendi, Moonseong Heo, Alain H. Litwin

**Affiliations:** 1Department of Psychology, College of Behavioral, Social, and Health Sciences, Clemson University, Clemson, SC 29634, USA; kaileib@clemson.edu (K.A.B.); ehammond@clemson.edu (E.H.); alain.litwin@prismahealth.org (A.H.L.); 2School of Health Research, Clemson University, Clemson, SC 29634, USA; 3Addiction Medicine Center, Prisma Health, Greenville, SC 29601, USA; erik.ortiz@prismahealth.org (E.G.O.); laksika.sivaraj@prismahealth.org (L.B.S.); ashley.king2@prismahealth.org (A.C.K.); 4Department of Medicine, Albert Einstein College of Medicine/Montefiore Medical Center, New York, NY 10461, USA; shadi.nahvi@einsteinmed.edu (S.N.);; 5Department of Health Promotion, Education, and Behavior, Arnold School of Public Health, University of South Carolina, Columbia, SC 29208, USA; thrasher@mailbox.sc.edu; 6Department of Medicine, University of South Carolina School of Medicine, Greenville, SC 29209, USA; scumby@email.sc.edu (S.C.); elig@email.sc.edu (E.G.); 7Department of Psychology, Universidad Loyola, 41704 Sevilla, Spain; 8Department of Public Health Sciences, Clemson University, Clemson, SC 29634, USA; mheo@clemson.edu

**Keywords:** buprenorphine treatment, opioid maintenance treatment, opioid use disorder, cognition, cognitive performance, cognitive impairment

## Abstract

People on buprenorphine maintenance treatment (BMT) commonly present cognitive deficits that have been associated with illicit drug use and dropout from buprenorphine treatment. This study has compared cognitive responses to the Stroop Task and the Continuous Performance Task (CPT) among individuals on BMT, with recent drug use, and healthy controls and explored the associations between cognitive responses and drug use, craving, and buprenorphine use among participants on BMT. The participants were 16 individuals on BMT and 23 healthy controls. All participants completed a 60 min laboratory session in which they completed the Stroop Task and the CPT, a saliva drug test, a brief clinical history that collected substance-use- and treatment-related information, and the Opioid Craving Scale. The results showed that the BMT participants presented more commission errors (MBMT participants = 2.49; Mhealthy controls = 1.38; *p* = 0.048) and longer reaction times (MBMT participants = 798.09; Mhealthy controls = 699.09; *p* = 0.047) in the Stroop Task than did the healthy controls. More days on buprenorphine were negatively associated with reaction time in the CPT (−0.52) and the number of commission errors (−0.53), simple reaction time (−0.54), and reaction time correct (−0.57) in the Stroop Task. Neither drug use nor craving was significantly associated with the results for the cognitive tasks. Relative to the control participants, the BMT individuals performed worse in terms of longer reaction times and more commission errors in the Stroop Task. Within the BMT participants, longer times on buprenorphine were associated with better cognitive results in terms of faster reaction times for both tasks and lower commission errors for the Stroop Task.

## 1. Introduction

Opioid use disorder (OUD) represents a public health problem that has steadily increased in the US since the 2000s [1,2]. In 2020, an estimated 2.7 million people aged 12 or older met the criteria for OUD in the US, and almost 10 million reported opioid misuse [3]. Increases in OUD have been associated with alarming increases in fatal and non-fatal overdoses [4,5]. Opioid misuse has also led to increases in infectious diseases associated with drug-injection behaviors, including HIV, acquired immunodeficiency syndrome (AIDS), and the hepatitis C virus [6]. Buprenorphine maintenance treatment (BMT) is widely used for treating OUD and has proven effectiveness [7] in reducing illicit drug use and the risk of overdose [8,9]. Unfortunately, ongoing illicit drug use, poor medication adherence, and dropout are common in buprenorphine treatment of OUD [10,11]. Therefore, it is important to study factors that may negatively influence treatment.

People with OUD frequently present cognitive impairment related to their opioid misuse [12,13,14], with the most pronounced deficits related to attention, executive functioning, and memory [15,16,17,18]. These cognitive impairments that people with OUD present have been demonstrated to have detrimental effects on engagement and retention in care [19]. Further, impairments in executive function have been linked to relapse in people with OUD [20].

People with OUD on BMT have continued to present cognitive deficits even while on stable doses of buprenorphine [21,22]. These cognitive deficits have been associated with illicit drug use and dropout from maintenance treatment programs among BMT patients [11,17]. Earlier studies have also shown that BMT patients generally present slightly worse cognitive outcomes relative to healthy controls [15,16,17,18,23]. It is important to highlight that the majority of studies involving BMT participants have excluded individuals with recent drug use [15,16,23,24], despite growing evidence indicating high rates of drug use among people with OUD and on buprenorphine [10,25], and therefore have limited the generalizability of their findings to real-world OUD treatment participants.

The purpose of this study was twofold: (1) to compare cognitive responses to the Stroop Task and the Continuous Performance Task (CPT) among individuals on buprenorphine treatment (i.e., BMT participants), with past 30-day illicit drug use, and healthy controls and (2) to explore the associations between cognitive responses to the Stroop Task and the CPT and OUD-related treatment outcomes, including drug use, cravings, and buprenorphine use, among participants on BMT.

## 2. Methods

### 2.1. Participants

Participants enrolled in a randomized clinical trial (RCT) testing the effectiveness of computer-assisted cognitive-behavioral therapy and recovery coaching (CBT4CBT + RC) versus treatment as usual (TAU) (NCT04824404) were invited to participate in this study. Details about this RCT have been published elsewhere [26]. The inclusion criteria for participating in the trial were (1) being 18 years of age or older; (2) having an OUD diagnosis; (3) being on buprenorphine maintenance for at least 30 days; and (4) having used any illicit substance within 30 days of screening, determined via self-report or a positive drug test at an office-based buprenorphine program. Participation in this laboratory study was optional for those enrolled in the clinical trial. A total of 16 participants of the RCT agreed to participate in this study and completed the laboratory session. The research data used in this study were collected at the beginning of the treatment before the delivery of any intervention. This study was approved by the Prisma Health institutional review board (IRB). All research activities were carried out according to the principles of the Declaration of Helsinki. Written informed consent was obtained from all participants.

A sample of 23 healthy controls were recruited at the Prisma Health Campus in Greenville, South Carolina. The healthy controls were matched based on gender and were eligible for participation if they had no history of or current mental health disorders or substance use disorders, which was determined using the Modified Mini Screen and the NIDA Drug Screen, respectively.

### 2.2. Measures

All participants completed all the assessments in a 60 min in-person laboratory session. Self-reported surveys were completed in REDCap, including those regarding sociodemographics and treatment- and drug-related measures. The Stroop Task and the CPT were administered via the CNS-Vital Sign computerized battery.

### 2.3. Sociodemographics

All participants completed a brief survey that collected basic sociodemographic information, including gender, age, marital status, sexual orientation, income, and education.

### 2.4. Treatment- and Drug-Related Measures

The BMT participants completed a brief clinical history that collected substance-use- and treatment-related information, including self-reported drug use, buprenorphine doses, and time on each dose. BMT participants also completed the Opioid Craving Scale (OCS) [27], a 3-item scale assessing urges, cue-induced-craving, and the likelihood of use on a visual analogue scale from 0 to 10. All participants completed a saliva toxicology screen for oxycodone, THC/cannabinoids, cocaine, opiates, methamphetamines, amphetamines, barbiturates, benzodiazepine, methadone, buprenorphine, and phencyclidine (ABMC, Kinderhook, NY, USA).

### 2.5. Cognitive Performance Tasks

Cognitive performance was evaluated with the Stroop Task and the CPT. The Stroop Task assesses response inhibition (ability to inhibit the production of an automatic response), selective attention (ability to maintain a cognitive or behavioral set in the presence of simultaneously available distracting or competing stimuli), and sustained attention (ability to maintain consistent behavioral responses throughout the duration of a task) across three parts. In the first part, four words printed in black (red, yellow, blue, and green) appeared randomly on a screen, and the participants were instructed to press the space bar as soon as the words were presented. During the second part, the words red, yellow, blue, and green were presented in color on the screen, and the participants were asked to press the space bar when the color of the word matched what the word said. During the third part, the same three colors would appear on the screen, printed in color, and the participants were asked to press the space bar when the color of the word did not match what the color said. During the CPT, which assesses sustained attention, a series of letters were presented randomly on a computer’s screen. The participants were asked to respond as quickly as possible to the letter “B” by pressing the space bar on the computer’s keyboard.

### 2.6. Procedures

All participants were scheduled for a single laboratory session. This laboratory session was conducted after baseline and prior to starting the 8-week intervention. After written informed consent was secured, the participants completed the brief sociodemographic survey. The BMT participants completed the clinical history and the OCS. Then, all participants completed both the cognitive tasks and the drug tests. After the study completion, each participant was paid 25 USD via ClinCards. All assessments were completed in an outpatient buprenorphine clinic located at the Prisma Health campus (Greenville, SC, USA). The same procedures were followed for both groups.

### 2.7. Data Analyses

Sociodemographic and drug-related characteristics were compared between the BMT participants and the healthy controls using *t*-tests for continuous variables and chi-squares for categorical variables. Differences between BMT individuals and healthy controls for the Stroop Task and the CPT were first assessed with the *t*-tests. Given that ages and education levels were different between groups, analyses were repeated through general linear models that controlled for age and education. Spearman’s correlations were conducted to explore the associations between cognitive tasks, self-reported drug use, saliva toxicology results, craving, buprenorphine doses, and time on each dose among the BMT participants. The confidence level for all analyses was 95%.

## 3. Results

Table 1 shows comparisons of sociodemographic and drug-related characteristics among the 16 individuals on BMT and the 23 healthy controls. The BMT sample was predominantly white (81.3%), female (62.5%), and an average of 37.5 years old, and 75% had at least some college education. The healthy control sample was mostly white (82.6%), female (69.6%), and an average of 28.9 years old, and 100% had at least some college education. The BMT participants were older (*p* = 0.01) and had lower educational attainment levels (*p* = 0.02) than the control group. With regard to treatment and drug-related characteristics, the BMT participants reported 11.5 days of drug use in the month prior to the study enrollment and, on average, a total daily buprenorphine dose of 19.3 and 314.7 days on buprenorphine.

General linear models that tested the differences between both groups for the Stroop Task and the CPT, adjusted for age and education, showed that the BMT participants presented higher commission errors (M_BMT participants_ = 2.49, SD = 0.4; M_healthy controls_ = 1.38, SD = 0.51; *F*(1,35) = 4.19, *p* = 0.048) and longer reaction times (M_BMT participants_ = 798.09, SD = 35.34; M_healthy controls_ = 699.09, SD = 45.02; *F*(1,35) = 4.25 *p* = 0.047) in the Stroop Task than did the healthy controls (see Table 2). The BMT participants did not differ from the healthy controls in the other two indices obtained from the Stroop Task, simple reaction time (*p* = 0.436) and complex reaction time (*p* = 0.062), or any of the CPT indices, including commission errors (*p* = 0.064), omission errors (*p* = 0.665), correct responses (*p* = 0.665), and choice reaction time correct (*p* = 0.186).

Spearman correlations between the cognitive tasks, self-reported drug use, saliva toxicology results, craving, buprenorphine doses, and time on each dose among the BMT participants are presented in Table 3. Twenty-seven correlations were generated between the CPT and Stroop Task indices, of which 17 were significant and positive. Duration of buprenorphine treatment was negatively correlated with reaction time (ms) for the CPT (−0.52) and with simple reaction time (ms) (−0.54), reaction time correcti (ms) (−0.54), and the number of commission errors (−0.53) in the Stroop Task. Finally, significant negative associations were found between all three OCS items (−0.53, −0.54, and −0.68) and having saliva toxicology tests that were positive for buprenorphine. The obtained correlation coefficients between craving and drug use with both cognitive tasks were not statistically significant (*p*s > 0.05).

## 4. Discussion

We found that our BMT patients performed worse in terms of longer reaction times and more commission errors in the Stroop Task compared to healthy controls. These results replicate those observed in earlier studies, which reported impaired cognitive performance among BMT participants relative to healthy controls. Specifically, those studies reported that BMT patients showed worse working memory, verbal memory, attention, and visual perception [15,16,17,18,23,24], BMT patients have also presented reduced cognitive function compared to healthy controls [15,17,28]. This is an important result given the link between cognitive function and treatment outcomes, including illicit drug use and dropout [11,17]. Future studies should explore whether adding a therapeutic component aimed at targeting and improving cognitive performance could improve cognitive function and impacts on buprenorphine retention and other OUD treatment-related outcomes.

This study found that among the BMT participants with OUD, longer times on buprenorphine were associated with better cognitive results in terms of faster reaction times for both tasks and fewer commission errors for the Stroop Task. Earlier studies conducted among BMT patients have shown that dose does not affect cognitive performance [17,21,22,29]. To our knowledge, this is the first study to show that a longer period of time on buprenorphine is associated with better cognitive outcomes. As noted by other authors, the pharmacological antagonism of the k-opiate receptor in buprenorphine may improve performance after opioid misuse by having an effect on and facilitating an optimal dopaminergic tone [15,17,28]. Therefore, it is possible that longer times on buprenorphine would result in better improvement of dopaminergic deficiency, which has been known to be crucial in cognitive functioning, in patients [28,30].

Our results also indicated that neither the results for the CPT nor the Stroop Task were significantly associated with self-reported craving for opioids or self-reported or objectively measured drug use. Our findings contrast with those of previous studies where craving and drug-seeking behaviors were associated with worse cognitive performance among people with OUD [21,31,32]. A potential explanation for our results may be related to the fact that these patients were on buprenorphine maintenance treatment for OUD. It is well-established that buprenorphine treatment helps reduce opioid craving and use [33,34]. It is possible that the impact of buprenorphine treatment on self-reported opioid craving and drug use may have lessened the association between cognitive performance and both craving and drug use. This lack of association highlights the need for further studies to replicate our study and determine whether there are individual factors that may determine the association between cognitive performance and opioid-related variables.

There are certain limitations of the current study that should be noted. First, the small sample size of this pilot study limited our statistical power and our ability to ascertain whether cognitive deficits were attributable to buprenorphine alone or buprenorphine and illicit drug use. Future studies should corroborate our findings with larger samples of BMT participants. Second, this study was a pilot cross-sectional study. It is important to explore whether cognitive functioning improves over time among BMT participants. Third, the sample was predominantly female, middle-aged adult, and white. Future studies should focus on recruiting more diverse and inclusive samples. Finally, the healthy controls were not age-matched with the BMT patients. While we controlled for age in the statistical analyses, there is potential for a confounding effect given the known negative relationship between age and cognitive functioning. This study also had strengths. First, the BMT participants enrolled in this study came from an outpatient maintenance clinic wherein illicit substance use is high, which mirrors the real-world reality wherein most patients present illicit drug use. Second, all our participants were screened in order to ensure that they were stabilized on their buprenorphine doses, therefore reducing variability.

In conclusion, this study has shown that cognitive performance, in terms of the results of the Stroop Task, is impaired among BMT patients with recent drug use relative to healthy controls. In addition, we evidenced that there is a positive correlation between time on buprenorphine and better cognitive results. Future research, with larger samples over longer periods of time and on different cognitive domains, will be needed to confirm our findings. We also suggest that these studies explore the relationship between cognitive performance and other OUD treatment outcomes, including adherence and retention. Taken together, our results indicate that cognitive therapy could be an appropriate strategy to improve the cognitive statuses of patients maintained on buprenorphine.

## Figures and Tables

**Table 1 ijerph-20-06610-t001:** Participants’ characteristics.

Characteristic	All Samples (*n* = 39)	BMT (*n* = 16)	HCs (*n* = 23)	
	M(SD)/*n* (%)	M(SD)/*n* (%)	M(SD)/*n* (%)	*p*
Sociodemographic Characteristics				
Age	32.4 (10.5)	37.5 (6.9)	28.9 (11.2)	0.01
Gender				0.74
Female	26 (66.7)	10 (62.5)	16 (69.6)	
Male	13 (33.3)	6 (37.5)	7 (30.4)	
Sexual Orientation				0.56
Heterosexual	36 (92.3)	14 (87.5)	22 (95.7)	
Other	3 (7.7)	2 (12.5)	1 (4.3)	
Race				1.0
White	32 (82.1)	13 (81.3)	19 (82.6)	
Other	7 (17.9)	3 (18.8)	4 (17.)	
Latino/Hispanic Ethnicity				0.21
Yes	7 (17.9)	1 (6.3)	6 (26.1)	
No	32 (82.1)	15 (93.8)	17 (73.9)	
Educational Attainment				0.02
<High School Graduate	4 (10.3)	4 (25.0)	0 (0.0)	
≥Some College	35 (89.7)	12 (75.0)	23 (100)	
Income				0.09
<15,000 USD	15 (38.5)	9 (56.3)	6 (26.1)	
≥15,000 USD	24 (61.5)	7 (43.8)	17 (73.9)	
Marital/Cohabitation Status				0.50
Single	25 (64.1)	9 (56.3)	16 (69.6)	
Other	14 (35.9)	7 (43.8)	7 (30.4)	
Hand Dominance				0.35
Right-Handed	32 (84.2)	11 (73.3)	21 (91.3)	
Left-Handed	3 (7.9)	2 (13.3)	1 (4.3)	
Ambidextrous	3 (7.9)	2 (13.3)	1 (4.3)	
Treatment- and Drug-Related Characteristics				
Saliva Positive for Drugs				
Any	9(23.1)	9(56.3)	0(0.0)	<0.001
Opioid/Opiates	1(2.6)	1(6.3)	0(0.0)	0.853
Stimulants	8(20.5)	8(50.0)	0(0.0)	<0.001
Benzodiazepine/barbiturate	2(5.1)	2(4.9)	0(0.0)	0.31
THC	2(5.1)	2(4.9)	0(0.0)	0.31
Polysubstances (≥2 drugs)	6(15.4)	6(37.5)	0(0)	0.006
Days Drugs Were Used Out of the Past 30		11.5 (10.8)	-	-
Opioid/Opiate Craving Scale Items				
Urges		1.7 (3.3)	-	-
Cue-Induced Craving		3.3 (3.6)	-	-
Likelihood of Use		4.3 (3.8)	-	-
Buprenorphine Daily Dose		19.3 (4.9)	-	-
Time on Buprenorphine (Days)		314.7(423)	-	-

Note. BMT = buprenorphine maintenance (patients); HCs = healthy controls.

**Table 2 ijerph-20-06610-t002:** Results for the Stroop Task and the CPT among healthy controls and participants on BMT.

Outcome	Crude Analysis Mean (SD)			Adjusted Analysis Est. (SE)			
	HC	BMT	*p*	HC	BMT	Diff (95% CL)	*p*
Continuous Performance Task							
Commission Errors	0.30 (0.47)	3.00 (5.19)	0.017	0.59 (1.28)	3.19 (1.01)	−2.6 (−5.35, 0.16)	0.064
Omission Errors	0.00 (0.00)	3.0 (8.87)	0.112	4.17 (1.98)	5.08 (1.55)	−0.92 (−5.18, 3.35)	0.665
Correct Responses	40.00 (0.00)	37.00 (8.87)	0.112	35.83 (1.98)	34.92 (1.55)	0.92 (−3.35, 5.18)	0.665
Choice Reaction Time Correct (ms)	419.70 (24.95)	485.31 (141.76)	0.035	446.21 (35.2)	496.52 (27.63)	−50.31 (−126.09, 25.46)	0.186
Stroop Task							
Commission Errors	0.78 (0.85)	2.38 (1.89)	0.001	1.38 (0.51)	2.49 (0.4)	−1.11 (−2.22, −0.01)	0.048
Simple Reaction Time (ms)	314.13 (41.89)	355.50 (119.04)	0.132	341.38 (31.29)	367.55 (24.56)	−26.17 (−93.52, 41.19)	0.436
Complex Reaction Time (ms)	599.87 (84.71)	639.63 (103.08)	0.195	548.16 (34.11)	618.06 (26.78)	−69.9 (−143.34, 3.54)	0.062
Reaction Time Correct (ms)	719.35 (111.82)	803.25 (124.50)	0.034	699.65 (45.02)	798.09 (35.34)	−98.44 (−195.36, −1.52)	0.047

**Table 3 ijerph-20-06610-t003:** Correlations between the Stroop Task, the CPT, self-reported drug use, saliva toxicology results, craving, buprenorphine doses, and time on each dose among BMT participants.

	1	2	3	4	5	6	7	8	9	10	11	12	13	14	15	16
*Continuous Performance Task*																
Commission Errors	-															
2. Omission Errors	0.44 **	-														
3. Correct Responses	−0.44 **	−1.00 *	-													
4. Choice Reaction Time Correct (ms)	0.31	0.45 **	−45 **	-												
*Stroop Task*																
5. Commission Errors—Stroop	0.20	0.39 *	−39 *		-											
6. Simple Reaction Time (ms)	0.38 *	0.37 *	−0.37 *	0.44 *	0.09	-										
7. Complex Reaction Time (ms)	0.26	0.27	−0.27	0.40 *	−0.06	0.47 **	-									
8. Reaction Time Correct (ms)	0.31	0.34 *	−0.34 *	0.30	0.00	0.50 **	0.80 **	-								
9. Buprenorphine Dose (mg)	−0.20	0.14	−0.14	−0.03	0.16	0.22	0.43	0.17	-							
10. Buprenorphine Time (Days)	−0.53 *	−0.40	0.40	−0.52 *	−0.53 *	−0.54 *	−0.27	−54 *	0.06	-						
11. Urges	−0.01	0.40	−0.40	0.28	0.29	0.32	−0.13	0.06	0.28	−0.23	-					
12. Cue-Induced Craving	0.09	0.24	−0.24	0.38	0.27	0.26	−0.03	0.17	0.32	−0.42	0.75 **	-				
13. Likelihood of Use	−0.33	−0.03	0.03	−00.3	0.22	0.05	−0.23	−0.17	−0.37	−0.18	0.81 **	0.68 **	-			
14. Self-Reported Days Drugs Used	−0.11	0.44	−44	0.21	0.03	0.24	0.16	0.34	0.40	0.132	0.25	0.01	0.04	-		
15. Saliva Positive for Any Drugs (Number)	0.11	0.13	−0.13	0.03	0.19	0.19	0.27	0.23	0.23	0.21	0.03	−0.06	−0.08	0.29	-	
16. Saliva Positive for Buprenorphine	0.00	0.15	−0.15	−0.25	−0.35	−0.19	0.09	−0.13	−0.07	0.27	−0.53 *	−0.54 *	−0.68 **	0.08	0.10	-

Note: * *p* < 0.05, ** *p* < 0.01.

## Data Availability

Data are available to qualified researchers upon reasonable request.

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
