# Peer review of "Preliminary Evidence of the Association between Time on Buprenorphine and Cognitive Performance among Individuals with Opioid Use Disorder Maintained on Buprenorphine: A Pilot Study"

_ijerph, 2023, doi:10.3390/ijerph20166610_

Round 1

Reviewer 1 Report

The author should clearly mention the number of individuals participating in the project. In participants section, they indicate that of 36 participants only 16 completed the study. And in the results, they indicate that 16 individuals were evaluated with BMT and 23 healthy patients... (indicate it clearly in the section of participants).

The authors do not indicate that the study was approved by the institutional ethics committee, also they should clearly indicate the bioethical considerations that were carried out throughout the study.

What is the novel contribution of this study?

The authors state that the objective of their study is to compare the cognitive response of individuals undergoing buprenorphine treatment with recent illicit drug use. Their study presents a very small sample size, and of the 16 individuals who were under buprenorphine treatment, only 9 of them showed evidence of being under any substance of abuse. The authors accept that the sample size is small and that it impacts the statistical power of the sample by limiting the ability to determine whether cognitive deficits are attributable to buprenorphine itself or to the combination with drugs of abuse. Although the study has the advantage that it corroborates with positive samples of drug use, it is important to increase the number of samples to be completely sure of the results and not speculate.

The quality of English is adequate.

Author Response

Reviewer 1

The author should clearly mention the number of individuals participating in the project. In participants section, they indicate that of 36 participants only 16 completed the study. And in the results, they indicate that 16 individuals were evaluated with BMT and 23 healthy patients... (indicate it clearly in the section of participants).

We concur with the Reviewer with regard to this issue. Based on her/his comment, we have modified the participants’ section to ensure that the number of participants is clear to our readers.

The authors do not indicate that the study was approved by the institutional ethics committee, also they should clearly indicate the bioethical considerations that were carried out throughout the study.

 We thank the reviewer for this comment. As suggested, we now indicate that our study was approved by the Prisma Health IRB, and all research activities were conducted according to the ethical principles of the Declaration of Helsinki.

What is the novel contribution of this study?

We thank the reviewer for this comment. The major innovation of the current study is that it included people with OUD who presented recent drug use. While most people undergoing buprenorphine treatment present recent illicit drug use, most studies exploring cognition among people with OUD on maintenance treatment excluded individuals with recent drug use.

The authors state that the objective of their study is to compare the cognitive response of individuals undergoing buprenorphine treatment with recent illicit drug use. Their study presents a very small sample size, and of the 16 individuals who were under buprenorphine treatment, only 9 of them showed evidence of being under any substance of abuse. The authors accept that the sample size is small and that it impacts the statistical power of the sample by limiting the ability to determine whether cognitive deficits are attributable to buprenorphine itself or to the combination with drugs of abuse. Although the study has the advantage that it corroborates with positive samples of drug use, it is important to increase the number of samples to be completely sure of the results and not speculate.

We thank the reviewer for this comment. All participants with OUD on buprenorphine (i.e., BMT participants) had recent drug use, defined as having used any illicit substance within 30 days of screening, determined either via saliva tox or self-report. With that being said, it is possible that only 9 of the 36 BMT participants had a positive drug test due to recent drug use may be detectable for up to five days.

Reviewer 2 Report

This study addresses cognitive deficits of patients on buprenorphine maintenance treatment (BMT), a common opioid effect that has been increasingly studied recently. .

The study comprises 16 individuals on BMT and 23 healthy controls. And concludes that BMT participants displayed more indicators of cognitive deficits than controls and more days on buprenorphine was negatively associated with was associated with better
cognitive results. Buprenorphine scores better than methadone in these measures. The manuscript is well written and comprehensively annotated with relevant references. The main issue is the small number of subjects, with adjusted p values and confidence intervals barely indicating significance. Nevertheless, the data corroborate earlier results and suggest a beneficial effect for buprenorphine on cognitive functions (tentatively attributed to antagonist activity at KOR). With this reservation, the study does contribute to the topic of opioid effects on cognitive functions.

Author Response

Reviewer 2

This study addresses cognitive deficits of patients on buprenorphine maintenance treatment (BMT), a common opioid effect that has been increasingly studied recently.

The study comprises 16 individuals on BMT and 23 healthy controls. And concludes that BMT participants displayed more indicators of cognitive deficits than controls and more days on buprenorphine was negatively associated with was associated with better cognitive results. Buprenorphine scores better than methadone in these measures. The manuscript is well written and comprehensively annotated with relevant references. The main issue is the small number of subjects, with adjusted p values and confidence intervals barely indicating significance. Nevertheless, the data corroborate earlier results and suggest a beneficial effect for buprenorphine on cognitive functions (tentatively attributed to antagonist activity at KOR). With this reservation, the study does contribute to the topic of opioid effects on cognitive functions. 

We thank the reviewer for his/her kind words. We firmly believe that this manuscript contributes to the current literature exploring the effects of opioids on cognition. We understand that the small sample size is a limitation of the current study; thus, we discussed and acknowledged this issue in the discussion section.

Reviewer 3 Report

one typo detected in 2.2 Measures section:  should be CNS Vital Signs, not CN-Vital Sign.

Author Response

Reviewer 3­­­

This brief report manuscript is focused on an important emergent aspect of addiction overall as well as specifically opioid use disorder (OUD) - cognitive impairment in this population and the relation of it to adherence and retention in treatment.

Irene Pericot-Valverde and colleagues presented cross-sectional results (one baseline visit that included featured assessments and a craving scale) related to cognitive performance on specific cognitive tasks vs. healthy controls from the randomized clinical trial where initial publication occurred in 2022 (Pericot-Valverde et al., 2022). This pilot sub-study was performed on a population of subjects who are on buprenorphine maintenance treatment (BMT) vs. healthy control. The portion of the subjects that went into this current analysis of buprenorphine maintenance subjects is n=16, and the healthy controls are n=23. Another important feature of this sample is that the BMT participants are those who are on BMT for at least 30 days, with ANY illicit drug use within the last 30 days. The authors also analyzed the duration of time that subjects were on BMT and their cognitive performance on the Stroop task and Continuous Performance Task (CPT). The authors highlight the fact that people who are on BMT continue to be cognitively impaired, especially those with recent drug use, and they concluded that the more time the subjects were on BMT the better their cognitive performance (results are within moderate size correlations).

We thank the reviewer for this thorough summary of our manuscript.

Abstract: Lines 20-22 the authors state (it seems prematurely) the conclusions from their study as "givens”. This calls into question as to why they have undertaken this study in the first place. In lines 27- 28, the authors also mention that in addition to declared cognitive tests study participants have completed" various self-report measures" that included "...treatment and drug-related measures". IT would be important here to be specific as to what domains of the addiction process these were related to, as being non-specific about these important assessments narrow down the study. It is notable that while craving assessment is mentioned in the Abstract, (and the results are present in Table 3) there is no discussion about the results anywhere in the Discussion section in the Abstract or in the Discussion in the paper itself, or Conclusions.

We apologize for these issues. The abstract section has been heavily edited based on the reviewer’s comment.

Introduction:

Lines 56-57: The introduction is positioned to state that the OUD population on BMT has cognitive impairments in attention, executive function, and memory, while in Lines 66-67, there is information that people who are on BMT are doing better cognitively than those on methadone. This BMT­ methadone theme is not getting further development in the text of the manuscript has no reflection on other parts of the manuscript, is briefly mentioned in the discussion, and is not all in the conclusions either. This study is not set up to do a comparison between BMT and methadone, therefore it would be advisable to delete methadone-related text, as it is not reflective of the study sample, its analyses, and conclusions. It would be beneficial to put the cognitive impairment in the OUD theme into a larger perspective and describe what are other domains of cognition (that haven't been explored in this study) and that those could be also related to clinically important outcomes.

We concur with the reviewer with regard to this issue. As suggested, we have removed the comparison between the BMT and methadone participants.

Lines 74-78: In the last paragraph of Introduction under (2), lines 77-78 there is a stated purpose to explore associations between the Stroop test and CPT in BMT subjects and CUD-related treatment outcomes: e.g., craving, drug use, buprenorphine use. Outside of the numerical results presented in Table 3, there is no interpretation given to those treatment outcomes and there is no text in the Discussion. It would be advisable to make the stated goals of this study to be concise and consistent with the results, discussion, and conclusions.

We apologize for this issue. We have revised the manuscript and now we provide the results related to craving and drug use in the results section. In addition, the new version of the manuscripts discusses the results related to the outcomes of drug use and craving.

Methods. Lines 95- 96: The healthy control group was matched only for gender, but not any other variables. It would have strengthened the study to do matching by the age range, especially in that further in the Discussion section the authors engage in the discussion regarding a potential benefit of improved dopaminergic tone with BMT. Age-related decline in dopaminergic function is well-known and can confound cognitive impairment, drug use, and affective vulnerability as well.

We concur with the reviewer with regard to this issue. We were unable to match the groups by age range. Therefore, we decided to control the statistical analyses by age. Based on the reviewer’s comments, we also included that we did not match by age range as a limitation of the current study.

Line 103: There is a typo there: CNS-Vital Signs battery, not CN-Vital Sign.

We thank the reviewer for this comment. This typo has been corrected.

Results: Lines 153-155: This study population looks restricted and not inclusive enough in regards to gender, age, and race. This makes the study results less generalizable to the overall OUD population.

We concur entirely with the reviewer that the sample could have been more diverse and inclusive. We now mention that most participants were female, white, and middle-aged adults as a limitation of the current study.

There is limited to none of the interpretation of findings on the clinically important variables related to craving, or any drug use. OCS scale is mentioned in 1 sentence (lines 181-182) without any interpretation of what this might mean clinically. This section needs to have a meaningful amount of clinical interpretation of the findings.

We kindly ask the reviewer to read the prior comment. We now present these results in the results section and discuss the meaning and implications in the discussion section.

Discussion: In this section, not connected with the rest of the paper, there is a notion on lines 195-197 that "...a therapeutic component may be needed" as a solution to improve cognitive function as well as OUD-related treatment outcomes. There is no further elaboration about this proposal that suddenly emerges in the discussion section.

We thank the reviewer for this comment.  This sentence has been modified based on the reviewer’s comment.

Further, in lines 200-204 authors are mixing the buprenorphine dose with the buprenorphine treatment duration, making an important statement that they are the first ones to show the connection between the treatment duration and improved cognitive performance. This needs to be corrected as the issue of the dose and treatment duration (time) are separate ones.

We agree that dose and treatment duration are different issues. Based on the reviewer’s comment, we have revised the sentence.

 Lines 204-209: comparison with methadone-maintained patients' cognitive performance might not be "legitimate", since this study has a relatively atypical patient population and it is not known whether this sample is a close match, at least in regards to the age range, to the studies authors cited and which is critical for dopaminergic tone discussion.

We concur with the Reviewer with regard to this issue. Based on her/his comment, we have removed this sentence.